# Induction of Chemerin on Autophagy and Apoptosis in Dairy Cow Mammary Epithelial Cells

**DOI:** 10.3390/ani9100848

**Published:** 2019-10-21

**Authors:** Bianhong Hu, Wenjuan Song, Yujie Tang, Mingyan Shi, Huixia Li, Debing Yu

**Affiliations:** 1College of Animal Science and Technology, Nanjing Agricultural University, Nanjing 210095, China; 2016105010@njau.edu.cn (B.H.); 2017105079@njau.edu.cn (W.S.); 2018105010@njau.edu.cn (Y.T.); 2College of Life Science, Luoyang Normal University, Luoyang 471934, China; smy2003@sina.com

**Keywords:** autophagy, apoptosis, Chemerin, bovine mammary epithelial cells

## Abstract

**Simple Summary:**

The process of mammary gland involution during the early is accomplished by both apoptosis and autophagy. Chemerin, a novel adipocytokine, plays a pivotal role in immune response and lipid metabolism, which was involved in the regulation of programmed cell death. This study focused on the relationship between autophagy and apoptosis in the presence of Chemerin in bovine mammary epithelial cells (BMECs). The results indicated that Chemerin could activate the complete autophagy process and induce apoptotic cascade in BMECs. The addition of Chloroquine (CQ), an inhibitor of autophagy, prompted Chemerin to have more obvious effects on apoptosis-related factors, which suggests that Chemerin-induced autophagy involves the intrinsic apoptotic pathway of BMECs. We found that the pro-apoptotic potential of Chemerin is further enhanced under conditions in which autophagy is effectively inhibited. Thus, exploring the role of autophagy and apoptosis on the involution of mammary epithelial cells will undoubtedly be of great importance in productivity improvement of dairy animals.

**Abstract:**

Involution of the mammary gland is a complex process controlled by various endocrine hormones and cytokine. As a novel adipocytokine, Chemerin not only plays a pivotal role in physiological and pathological processes such as immune response and lipid metabolism, but is also involved in the regulation of programmed cell death, including autophagy and apoptosis. The purpose of the present study was to elucidate whether autophagy and apoptosis of bovine mammary epithelial cells (BMECs) was triggered by Chemerin. BMECs were cultured and treated with Chemerin in vitro. The expression of autophagosome-forming marker, microtubule-associated protein 1 light chain 3 II (LC3-II) and sequestosome-1 (SQSTM 1, best known as p62), a substrate of autophagosome degradation were detected. The result showed that Chemerin significantly decreased the expression of p62 and markedly induced the conversion of LC3-I to LC3-II. The ratio of Bcl2-associated X and B-cell lymphoma-2 (Bax/Bcl-2) and the activity of caspase-3 were up-regulated after being treated by Chemerin, and the apoptotic rate was also significantly increased. These results suggested that Chemerin promoted the occurrence of autophagy and apoptosis in BMECs. Chloroquine (CQ), which is an inhibitor of autophagy. To explore effects of Chemerin on apoptosis, we prevented Chemerin-induced autophagy by pre-adding CQ in BMECs. Interestingly, this part of the experiment helped us find that all effects of Chemerin on apoptosis of BMECs could be enhanced with the inhibition of autophagy. Our study demonstrates that Chemerin-induced autophagy and apoptosis are mutually regulated in BMECs, but the specific mechanism remains to be further researched.

## 1. Introduction

Mammary gland involution is an important physiological process for subsequent lactation. Apoptosis is the primary pattern of self-repair of mammary epithelial cells in involution stages [1,2,3,4]. However, there are also a lot of cells exhibiting morphological features of autophagy during involution [5]. Whether the involution of the mammary gland is normal or not will directly affect the collapse of the mammary duct and acinar structure and lactation performance. The unique physiological synthesis and secretion functions of mammary epithelial cells enable them to be used as an ideal model for the study of mammary gland bioreactor [6,7]. Thus, exploring the role of autophagy and apoptosis on the involution of mammary epithelial cells will undoubtedly be of great importance in productivity improvement of dairy animals. 

Autophagy is a self-stabilizing mechanism prevalent in eukaryotic cells [8,9]. Autophagic activity is known to be regulated by a complex array of signals. When autophagy is induced, microtubule-associated proteins 1A/1B light chain 3B (MAP1LC3B, best known as LC3) is coupled to autophagosomes and converted from LC3-I to LC3-II, which has been used as a recognized indicator for autophagy evaluation and assay [10]. As another indicator of autophagy detection, the multifunctional protein p62 can recruit ubiquitinated proteins through the ubiquitin-linking domain to form oligomers that interact with LC3 and eventually it is degraded by lysosomes [11]. Autophagy is observed at all stages of the lactation cycle and attain peak yield during the dry period [12]. Apoptosis, a process of cell death, is co-powered by multiple genes. Mammary epithelial cell apoptosis is an integral component of tissue remodeling [1]. Interestingly, the morphological changes of apoptosis also exist in acinar cells [5]. Nevertheless, the effect of autophagy and apoptosis remains controversial in the process of cell survival. Studies have reported that autophagy provides nutrients to catabolize apoptosis through catabolism, prolonging the survival of cells during metabolic stress [13,14], but studies have also confirmed that autophagy can act as a mechanism of death to synergistically promote apoptosis [15]. Therefore, the specific biological effect exerted by autophagy is to match the physiological environment in which the cells are located.

The involution of the mammary gland is a physiological and biochemical process involving the interactions of various hormones and cytokines. Chemerin is a novel adipokine that is identified as a natural ligand for ChemR23 with clear immune function [16]. Recently, studies have shown that Chemerin relieves the pathological state of metabolically hypertensive rats by regulating autophagy [17]. Chemerin induces granulosa cell apoptosis by promoting caspase-3 activation to increase DNA fragmentation in vivo [18]. In conclusion, Chemerin is closely related to apoptosis and autophagy when it exerts the biological function. Nevertheless, the effect of Chemerin on autophagy in bovine mammary epithelial cells (BMECs) is largely unknown. The purpose of this study is to provide some prerequisites for further researches on the molecular mechanism of autophagy and apoptosis of BMECs.

## 2. Materials and Method

### 2.1. Chemicals and Reagents

The recombinant protein Chemerin (2325-CM-025), this is the mouse Chemerin, which was obtained from R&D Systems (MN55413, Minneapolis, MN, USA). Bax (50599-2-1g), Bcl-2 (12789-1-AP), p62/SQSTM1 (18420-1-AP), and GAPDH (10494-1-AP) antibodies were obtained from Proteintech Group Inc (Chicago, IL, USA). Caspase-3 (bs-0081R) was purchased from Bioss Biotechnology Co., Ltd. (Beijing, China). LC3B (NB100-2220SS) was obtained from NOVUS Biologicals (Littleton, Colorado, CO, USA). Mouse keratin-18 monoclonal (Cytokeratin18) (ab52459) was purchased from Abcam (Cambridge, MA, USA). FITC-labeled goat anti-rabbit IgG (FMS-Rbaf4701) were obtained from Fcmacs Biotech Co., Ltd. (Nanjing, Jiangsu, China). Apoptosis detection kit (556454) was obtained from BD Biosciences (San Jose, California, CA, USA). Chloroquine diphosphate (C6628) was obtained from Sigma-Aldrich (St Louis, MO, USA).

### 2.2. Cell Culture

BMECs were a gift from Dr. Sun Youping (Harvard University). The cells were maintained in DMEM/F12 complete medium containing 10% fetal bovine serum (FBS; Sigma-Aldrich, St. Louis, MO, USA) and incubated at 37 °C in a humidified atmosphere of 5% CO_2_. Digestion and subculture were carried out when the cells were grown up to 90% confluence in DMEM/F12 medium (DMEM/F12, Biological Industries, Beit Haemek, Israel).

### 2.3. Experimental Design and Treatment

To assess the role of Chemerin on autophagy and apoptosis, BMECs were incubated with Chemerin (0.1 µg/mL) for 24 h and divided into control group and Chemerin group. The selection of Chemerin concentration and treatment time was mainly based on previous research by our team, which confirmed that 0.1μg/mL Chemerin treatment for 24 h had the lowest cytotoxicity.

For other analysis, cells were randomized into four groups: (1) Control group, (2) Chemerin group (cells were cultured with Chemerin (0.1 µg/mL) for 24 h), (3) CQ group (cells were pre-incubated with CQ (10 µmol/L) for 2 h and then cultured in normal medium), and (4) Chemerin + CQ group (cells were cultured with CQ (10 µmol/L) for 2 h and then exposed to Chemerin (0.1 µg/mL) for 24 h). The cells were cultured to collect and perform each analysis after the indicated time.

### 2.4. Quantitative Real-Time PCR

Cells in the logarithmic growth phase with a fusion rate of 80% were treated at the specified time, and total RNA was extracted using Trizol (Invitrogen, Carlsbad, CA, USA) after the treatment. cDNA was synthesized using a Prime ScriptTM RT Master Mix enzyme (TaKaRa Biotechnology, Co., Ltd., Japan) and RNA expression level were assessed by using SYBR^®^ Premix Ex TaqTM (TaKaRa Biotechnology, Co., Ltd., Japan). The sequences of primer are listed in Table 1. Gene expression data were normalized to the glyceraldehyde-3-phosphate dehydrogenase (*GAPDH*) and the results were analyzed using a 2^−△△ct^ model. 

### 2.5. Western Blot Analysis

The cells of each treatment group were thoroughly washed with pre-cooled phosphate buffer saline (PBS). Cells were placed on ice for 30 min and added 300ìL of Radio Immunoprecipitation Assay (RIPA, (Beyotime Biotechnology, Shanghai, China) lysis buffer containing 1% phenylmethylsulfonyl fluoride (PMSF (Solarbio Science and Technology Corporation, Beijing, China) to extract the proteins. The lysate was transferred to the centrifuge tube and centrifuged at 12,000× g for 15 min at low temperature. The total protein concentration in the supernatant was estimated by the Bradford assay (Beyotime, Shanghai, China). After high temperature denaturation, the equal amounts of protein were separated by 12% SDS-PAGE (GenScript Biotech Corporation, Nanjing, China) (140 V, 90 min) and electrotransferred (90 V, 90 min) to the PVDF (merckmillipore LtD, New York, USA) membranes. After transfer, the membranes were incubated with 5% skim milk for 1 h at room temperature, then rabbit polyclonal antibodies including Bax, Bcl-2, caspase-3, LC3B, and p62/SQSTM1 (1:1000) were added and kept at 4 °C overnight. After thorough washing, the membranes were incubated with HRP-conjugated secondary antibody (1:6000) for 2 h at room temperature. Finally, detection was performed by ECL (Biosharp, Hefei, China) and densitometry was performed by using Image J software (National Institutes of Health, Bethesda, MD, USA).

### 2.6. Immunofluorescence Staining

The 24-well culture plate was placed with climbing slices laid ahead in the super-clean bench and irradiated with ultraviolet light for 30 min to achieve sterility. The cells were evenly inoculated on the sterile 24-well culture plate with climbing slices laid ahead, when the degree of cell fusion reached 80%, the corresponding treatment was carried out. After the treatment, the cell morphology was fixed by addition of 4% paraformaldehyde for 30 min. After washing, the cells were permeabilized with 0.5% Triton X-100 (Solarbio Science and Technology Corporation, Beijing, China) for 15 min and then transferred to 5% BSA for 1 h. Subsequently, the cells were labeled with primary antibody (1:500 dilution) overnight at 4 °C, and a 1:500 dilution of the fluorescence-labelled secondary antibody was added at room temperature for 1 h. DAPI was subjected to nuclear staining for 15 min in the dark, and fluorescence visualized using a microscope (Olympus, Tokyo, Japan). The average fluorescence intensity was measured in three randomly selected visual fields using Image J software (National Institutes of Health, Bethesda, MD, USA).

### 2.7. Flow Cytometry (FCM)

Apoptosis detection kit was employed to the measured apoptosis of BMECs. Briefly, the cultured cells in growth medium were collected from each treated group, transferred to a dedicated flow tube and washed with PBS. Subsequently, the cell pellet obtained by centrifugation was suspended in 100μL of binding buffer. The binding buffer is a 10× concentrate composed of a 0.2 µm sterile filtered 0.1 M Hepes (pH 7.4), 1.4 M NaCl, and 25 mM CaCl_2_ solution, it was diluted by PBS at a ratio of 1:20 before use. 5μL of annexin V-PE/7-ADD and propidium iodide (PI) were added for 15 min to distinguish cell status, and the number of early apoptotic cells was counted by flow cytometry.

### 2.8. Statistical Analysis

All data were expressed as mean ± SEM. Differences between groups were analyzed by analysis of variance, followed by Duncan’s multiple comparisons test. Statistical significance was shown as *p*< 0.05.

## 3. Results

### 3.1. Identification of BMECs

Cytokeratin-18 belongs to the family of epithelial intermediate filament proteins, which forms the cytoskeleton of epithelial tissues. In this assay, BMECs were identified by immunofluorescence staining of cytokeratin-18. As shown in Figure 1, the cells grown to confluency, which formed a monolayer and aggregated with morphological characteristics of cobblestones as well as the culture samples contained 95% of cytokeratin-18 positive cells. It was proven that the cells were bovine mammary epithelial cells.

### 3.2. Chemerin Promotes Autophagy and Induces Apoptosis in BMECs

Among the many factors involved in the regulation of autophagy, LC3 is a marker protein of autophagosome formation. When the cells receive the autophagy induction signal, LC3-I present in the cytoplasm will enzymatically cleave a small segment of the polypeptide and transform into LC3-II involved in the extension of the membrane. To analyze the effects of Chemerin on the autophagy in BMECs, cells were treated with 0.1μg/mL recombinant protein Chemerin for 24 h. The LC3-labeled BMECs were visualized by confocal microscopy. As shown in Figure 2A,B, the green fluorescent protein were more easily seen in the Chemerin group compared to the control group (*p* < 0.05), indicated that Chemerin involved in the regulation of autophagy and promoted the formation of autophagosome structures. 

With the gradual comprehensive understanding of autophagy, the viewpoint that the pattern of high LC3-II low p62 is the gold indicator of autophagy that has been widely accepted. To better assess the influence of Chemerin on autophagy, the expression of *LC3* and *p62* at the transcriptional (Figure 2C) and translational (Figure 2D–E) levels were examined, respectively. The data showed that the Chemerin treatment markedly increased the protein expression of LC3-II compared to the control group (*p* < 0.01). Meanwhile, there was a clear decrease of the mRNA and protein expression of *p62* after 24 h exposure to Chemerin (*p* < 0.05). These data displayed that Chemerin induced autophagy in BMECs.

Next, we determined the apoptotic rate by Annexin V-PE/7-ADD staining to evaluate the effect of Chemerin on the apoptosis of BMECs. As shown in Figure 3A,B, Fluorescent dot plots located in the upper right quadrant and the lower right quadrant correspond to cell populations in the late stage and early stages of apoptosis, respectively. After treatment with Chemerin for 24 h, cells showed a certain proportion of apoptosis and most of cells were in the upper right quadrant (*p* < 0.05), indicating that Chemerin can induce late apoptosis in BMECs. 

The occurrence of apoptosis involves the activation of a series of molecular events. Bcl-2, a survival protein, can bind to pro-apoptotic protein Bax to inhibit apoptosis. Caspase-3 is the ultimate performer of apoptosis. In further study, we evaluated the expression of a series of apoptosis-related genes after adding Chemerin. The quantitative results showed that the addition of Chemerin in the cells significantly enhanced the *Bax/Bcl-2* ratio (*p* < 0.01) (Figure 3C), which was consistent with the immunoblot results (*p* < 0.01). In addition, the protein level of activated caspase-3 was also markedly elevated (*p* < 0.05) (Figure 3D,E). These data indicated that Chemerin promotes apoptosis of BMECs by regulating key genes for apoptosis.

### 3.3. CQ Has an Inhibitory Effect on Autophagy Induced by Chemerin

As a classical inhibitor of autophagy, CQ can destroy the normal function of lysosomes to inhibit the degradation of autophagic substrates. To explore the role of autophagy in Chemerin-induced apoptosis, BMECs were incubated with 10 μmol/L CQ for 2 h before exposure to 0.1 μg/mL Chemerin for 24 h. Immunofluorescence assay illustrated that Chemerin reduced the expression of p62 protein (*p* < 0.01), but CQ pretreatment significantly reversed this phenomenon (*p* < 0.05), indicating that CQ blocked Chemerin-induced degradation of autophagy substrates (Figure 4).

To further determine the role of CQ on autophagy induced by Chemerin, we next examined the expression of autophagy crucial genes at a molecular level. It can be observed from Figure 4c–e that the mRNA expression and protein concentration of *p62* was remarkably downregulated after Chemerin treatment (*p* < 0.05), but CQ suppressed this stimulative effect of Chemerin (*p* < 0.05). Similarly, pretreatment with CQ also enhanced LC3-II protein expression in the Chemerin group (*p* < 0.05), suggested that CQ could inhibit the autophagosome fusion with lysosomes and lead to accumulation of autophagosomes. Taken together, these results demonstrated that CQ could effectively block Chemerin-induced autophagy and can be used for subsequent experiments. 

### 3.4. Inhibition of Autophagy Enhances Chemerin-Induced Apoptosis in BMECs

Autophagy and apoptosis regulate each other during cell death, which may be triggered by common upstream signals. To evaluate whether Chemerin-induced autophagy affects apoptosis in BMECs, apoptosis rates were measured after distinguishing the cell status with Annexin V-PE and 7-ADD. We found that Chemerin induced a significantly apoptosis rate compared with the control group (*p* < 0.05), and this phenomenon was aggravated by treatment combined with CQ (*p* < 0.05) (Figure 5A,B), suggesting that CQ pretreatment increased the apoptosis induced by Chemerin in BMECs.

To explore the molecular mechanism by which Chemerin regulates apoptosis in the context of autophagy inhibition by CQ, we examined the expression of several apoptosis-related genes. Quantitative results showed that the mRNA ratio of *Bax/Bcl-2* in BMECs was significantly upregulated after Chemerin treatment (*p* < 0.05) (Figure 5C). In addition, we further examined the protein expression of apoptotic factors by Western blot (Figure 5D,E), the results showed that the concentrations of Bax/Bcl-2 and cleaved-caspase-3 were similarly up-regulated (*p* < 0.05). Interestingly, CQ treatment further promoted the effect of Chemerin on apoptosis-related genes in BMECs (*p* < 0.05). The above results revealed that inhibition of autophagy further promotes Chemerin-induced apoptosis.

## 4. Discussion

Recent research has revealed that the process of mammary gland involution during the early years is accomplished by both apoptosis and autophagy [19,20]. In addition to the regulation of endocrine hormones, local cytokines also become the focus of research on the autophagy and apoptosis during mammary gland involution. As a newly identified adipokine, Chemerin binds to CMKLR1 to amplify the lactation performance of bovine mammary epithelial cells [21]. As a novel adipocytokine, Chemerin not only plays a pivotal role in physiological and pathological processes such as immune response and lipid metabolism, but is also involved in the regulation of programmed cell death, including autophagy and apoptosis. The purpose of the present study was to elucidate whether autophagy and apoptosis of bovine mammary epithelial cells (BMECs) was triggered by Chemerin. The focus of this study was to explore the relationship between autophagy and apoptosis in the presence of Chemerin in BMECs. We found that the pro-apoptotic potential of Chemerin was further enhanced under conditions in which autophagy was effectively inhibited. 

Autophagy is a self-catabolic process in eukaryotes that relies on the autophagy-lysosomal pathway to degrade damaged or misfolded proteins [14]. Chemerin could activate autophagy in C2C12 myotubes via Akt/FoxO3α-dependent signaling pathway [22]. To determine whether Chemerin activated autophagy in BMECs, the cells were exposed to culture medium containing 0.1 μg/mL Chemerin for 24 h. Immunofluorescence assay showed that Chemerin exposure enhanced the protein expression of LC3-II. As a recognized indicator of autophagy evaluation and assay, LC3 exists in two forms in cells. LC3-I is covalently linked to phosphatidylethanolamine (PE) by processing of a ubiquitin-like lipid coupling system to form LC3-II and inserted into the bilayer membrane of autophagosomes [10]. Thus, the content of LC3-II can be used to assess the level of autophagy. As the research of autophagy progresses deeper, it is generally considered that the increase or decrease of autophagosome formation does not essentially reflect the level of autophagy. As a type of adaptor protein, p62 can be ligated to ubiquitinated protein aggregates at one end, and the other end is specifically linked to autophagosome membrane LC3 and degraded in autophagosomes to become a marker of autophagic flux completion [11]. Thus, high LC3-II/low p62 pattern may be indicative of activation of intact autophagy. In our current study, we observed Chemerin stimulation significantly induced LC3-I to LC3-II protein conversion, and down-regulated p62 expression at both transcriptional and translational levels. The results were similar to the role of Licarin A (LCA) in lung cancer cell lines cultured in vitro [23]. The results indicated that Chemerin could activate the complete autophagy process in BMECs.

The mammary gland forms apoptotic bodies by DNA fragmentation and chromatin condensation to reduce the number of bovine mammary epithelial cells after weaning [19]. Cell apoptosis is an initiative cell death under a series of genes regulation. The protein balance of pro-apoptotic Bax and anti-apoptotic Bcl-2 decides the fate of the cell, and may induce cytochrome c release [24]. Cytochrome c released from mitochondria into the cytoplasm activates downstream caspase-3, which ultimately performs cell death [25,26]. In this experiment, we observed that the ratio of Bax/Bcl-2 significantly increased after Chemerin treatment, and the protein concentration of cleaved-caspase-3 was also up-regulated. The results were consistent with our previous studies in ovarian granulosa cells from cows (data not shown). Those results indicated that Chemerin induced apoptotic cascade in BMECs.

There are many common upstream pathways between autophagy and apoptosis [27]. Furthermore, autophagy and apoptosis can regulate each other, autophagy can be used as a trigger to induce apoptosis [28], autophagy can also antagonize apoptosis [29]. CQ is the only autophagy inhibitor that has been certified to be safely used in vivo and in vitro to target lysosome [30], which blocks the process of autophagosome lysosomal fusion and metabolism by modifying the lysosomal pH [31]. It has been demonstrated that the use of CQ prevents the scavenging effect of p62 on abnormal aggregated proteins, thereby inhibiting the flux of autophagy [32], which is consistent with our results that CQ pretreatment increased p62 fluorescence intensity. In further studies we observed a significant increase in *p62* expression at mRNA and protein levels after CQ exposure. In addition, the expression of LC3-II protein was also up-regulated. This is consistent with previous work in human umbilical vein endothelial cells, in which CQ promoted the accumulation of autophagosomes [23]. Taken together, our data support the view that treatment of CQ does not affect the formation of autophagosome but prevents the subsequent degradation process. Interestingly, the addition of CQ prompted Chemerin to exert more obvious effect on apoptosis-related factors, which suggests that Chemerin-induced autophagy involves the intrinsic apoptotic pathway of BMECs. 

## 5. Conclusions

In summary, these findings proved that apoptosis and autophagy interacted with each other as a partner to induce Chemerin-treated BMECs death, and the function of Chemerin to promote cell apoptosis would be significantly enhanced if the autophagy was suppressed, and the simple regulatory roadmap as shown in Figure 6. BMECs were treated with the Chemerin, which induced autophagy through promoting degradation of p62 and conversion of LC3I to LC3II and also induced apoptosis by upregulation the Bax/Bcl-2 and activated caspase-3. In addition, there was a complex interplay between apoptosis and autophagy under the Chemerin treatment, CQ pretreatment could effectively inhibit autophagy and aggravate apoptosis.

## Figures and Tables

**Figure 1 animals-09-00848-f001:**
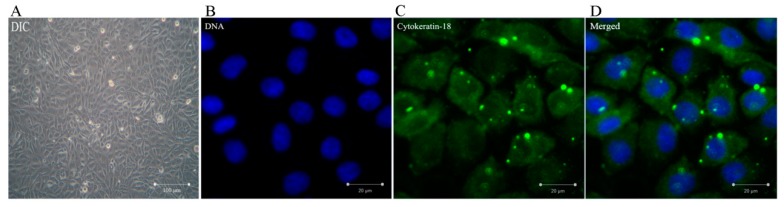
The culture of bovine mammary epithelial cells (BMECs) and the immunofluorescence of cytokeratin-18 in vitro. It was proven that the cells were bovine mammary epithelial cells. (**A**) Morphology of BMECs. (**B**) The BMECs were inoculated into the cell slide, and the keratin 18 immunofluorescence staining was performed. A green positive reaction was observed around the blue nucleus. Detection of nuclei by 4’, 6-diamidino-2-phenylindole (DAPI) staining (blue). (**C**) Detection of cytokeratin-18 protein (green) by immunofluorescence staining. (**D**) Merged picture of (**B**) and (**C**).

**Figure 2 animals-09-00848-f002:**
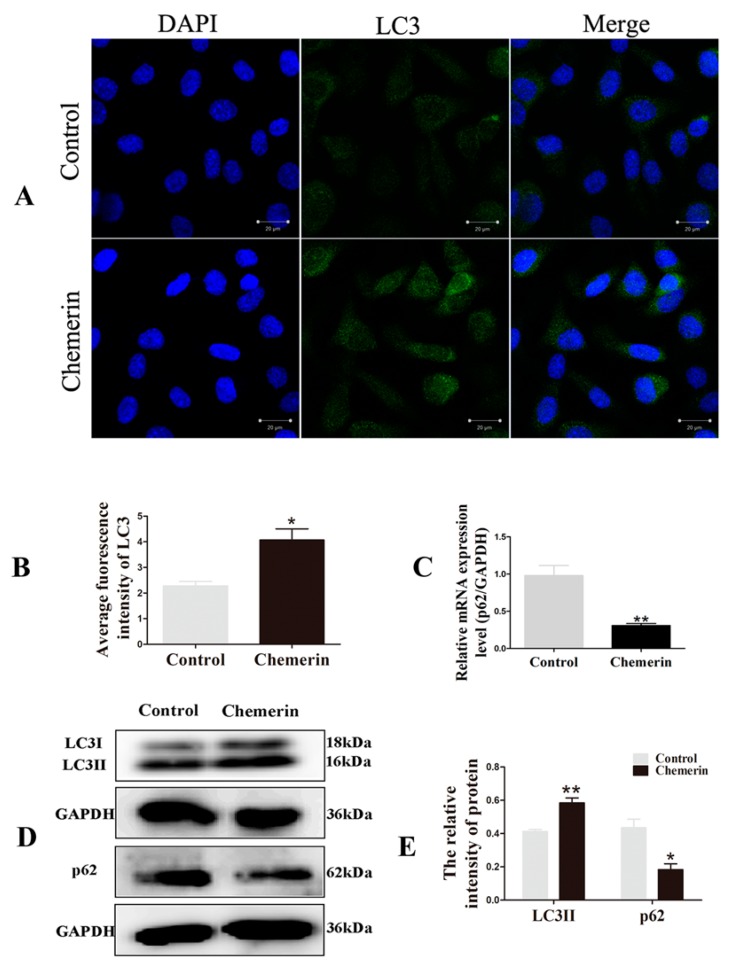
Chemerin promotes autophagy of BMECs. Cells were exposed to the culture medium with 0.1μg/mL Chemerin form 24 h. (**A**) LC3 protein distribution was detected by immunofluorescence staining. (**B**) The average fluorescence intensity of LC3 was determined in three randomly selected fields of view by Image J software. (**C**) Messenger RNA (mRNA) expression of sequestosome-1 (*SQSTM 1*, best known as *p62*) profiling. (**D**) Protein expression of LC3 and p62 was evaluated by Western blot. (**E**) The fluorescence intensity of protein was assayed by Image J software. All data were expressed as means ± SEM, with each treatment performed in triplicate. A single asterisk indicates a statistical difference (*p* < 0.05), and a double asterisk indicates a statistical difference (*p* < 0.01) when compared with control.

**Figure 3 animals-09-00848-f003:**
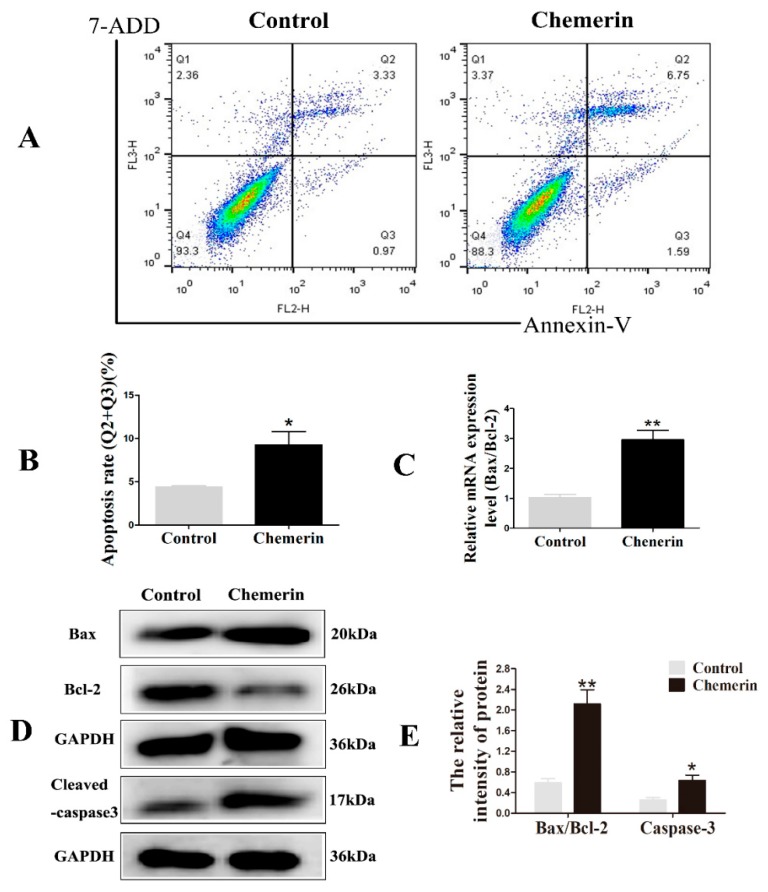
Chemerin-induced BMECs apoptosis in BMECs. Cells were stimulated with Chemerin for 24 h. (**A**) Analysis of apoptosis via flow cytometry. (**B**) Percent apoptosis was determined in Q2 + Q3. (**C**) The ratio of Bcl2-associated X and B-cell lymphoma-2 (*Bax/Bcl-2*) was analyzed by qRT-PCR. The real-time PCR results were analyzed using a 2^−△△ct^ model. (**D**) Protein expression of Bax/Bcl-2 and cleaved-caspase-3 were evaluated by Western blot. (**E**) Image J software was used to quantify the apoptosis-related protein levels. All data are expressed as means ± SEM, with each treatment performed in triplicate. A single asterisk indicates a statistical difference (*p* < 0.05), and a double asterisk indicates a statistical difference (*p* < 0.01) when compared with the control.

**Figure 4 animals-09-00848-f004:**
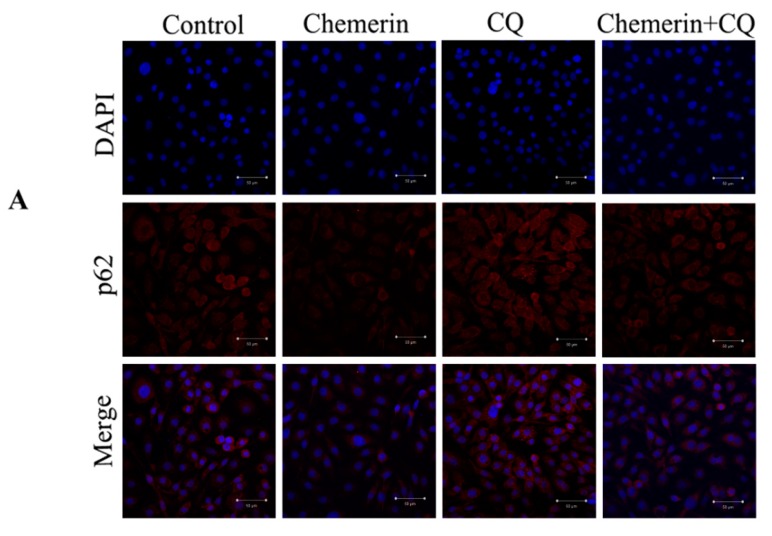
Chloroquine (CQ) inhibits Chemerin-induced autophagy in BMECs. The BMECs were treated with 10 μmol/L CQ for 2 h and then incubated with 0.1 μg/mL Chemerin for 24 h. (**A**) p62 protein expression was observed by confocal microscopy in different treatment groups. (**B**) The average fluorescence intensity of p62 was quantified in different treatment groups. (**C**) mRNA expression of *p62* was detected by qRT-PCR in different treatment groups. The real-time PCR results were analyzed using a 2^−△△ct^ model. (**D**) LC3 and p62 expression were analyzed by Western blot in different treatment groups. (**E**) Densitometric analysis of proteins in (**D**) by Image J software. All data are expressed as means ± SEM, with each treatment performed in triplicate. A single asterisk indicates a statistical difference (*p* < 0.05), and a double asterisk indicates a statistically significant difference (*p* < 0.01) when compared with the control. A single ampersand indicates a statistical difference (*p* < 0.05), and a double ampersand indicates a statistical difference (*p* < 0.01) when compared with the Chemerin-treatment group.

**Figure 5 animals-09-00848-f005:**
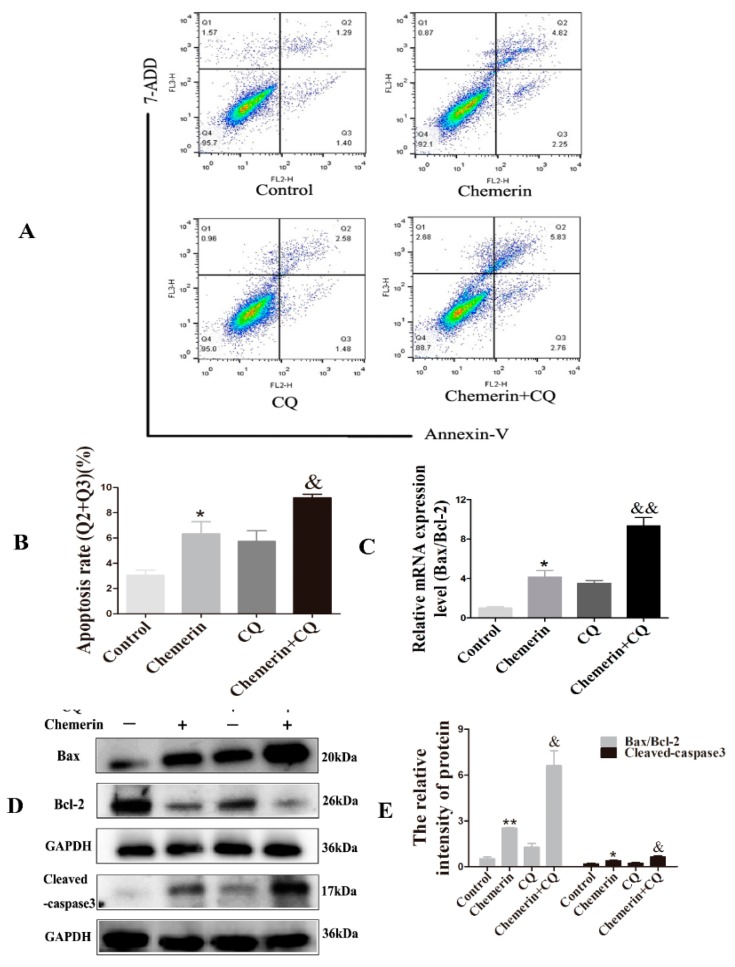
**Inhibition of autophagy promotes apoptosis induced by Chemerin in BMECs.** Cells were incubated in CQ-containing medium for 2 h before exposuring to the medium with 0.1 μg/mL Chemerin for 24 h. (**A**) Apoptosis was detected by Annexin V-PE/7-ADD staining. (**B**) Cells apoptosis rate were analyzed in Q2 + Q3. (**C**) mRNA expressions of *Bax/Bcl-2* were detected by qRT-PCR analysis. The real-time PCR results were analyzed by a 2^−△△ct^ model. (**D**) The protein expression of Bcl2-associated X and B-cell lymphoma-2 (Bax/Bcl-2) and cleaved-caspase-3 were examined by Western blot. (**E**) Quantification of the apoptosis-associated proteins were analyzed in (**D**). All data were expressed as means ± S.E.M, with each treatment performed in triplicate. A single asterisk indicated a statistical difference (*p* < 0.05), and a double asterisk indicated a statistical difference compared with control (*p* < 0.01). A single ampersand indicated a statistical difference (*p* < 0.05), and a double ampersand indicated a statistical difference when compared with the Chemerin-treatment group (*p* < 0.01).

**Figure 6 animals-09-00848-f006:**
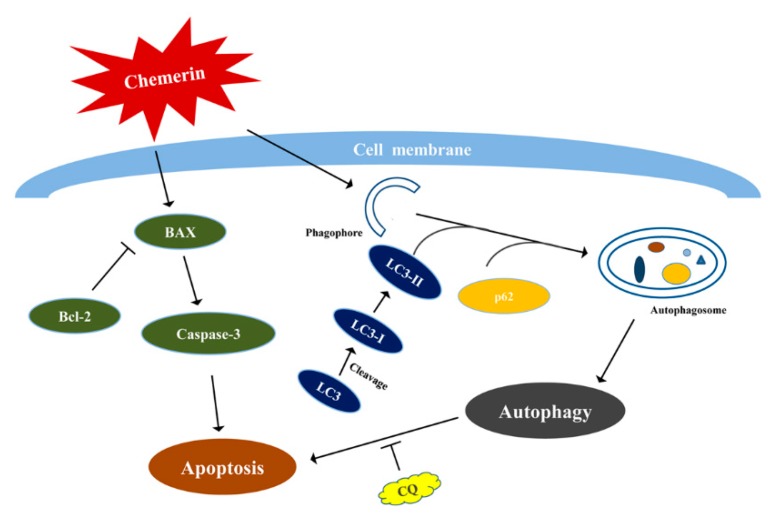
A schematic illustration of the role of Chemerin in BMECs. Chemerin induced autophagy through promoting degradation of p62 and conversion of LC3I to LC3II. Chemerin also induced apoptosis by up-regulation of the Bax/Bcl-2 and activating caspase-3. In addition, there was a complex interplay between apoptosis and autophagy under Chemerin treatment, CQ pretreatment could effectively inhibit autophagy and aggravate apoptosis.

**Table 1 animals-09-00848-t001:** Primer sequences for qRT-PCR.

Genes	Forward (5′–3′)	Reverse (5′–3′)
*p62*	ATTGAGCCAGCTCAGGCTGT	CTGGCTGGAAGTCAGGCTGT
*Bax*	CCAGCAAACTGGTGCTCAAGG	AGCCGCTCTCGAAGGAAGTC
*Bcl-2*	AGCATCGCCCTGTGGATGAC	CAGCCTCCGTTGTCCTGGAT
*GAPDH*	AAGGTCGGAGTGAACR	CGTTCTCTGCCTTGACTGTG

*P62*: Sequestosome 1, an autophagy selective substrate. *Bax*: BCL2-Associated X Protein, plays a role in the mitochondrial apoptotic process. *Bcl-2*: B-cell lymphoma-2, a cancer gene that inhibits apoptosis. *GAPDH*: glyceraldehyde-3-phosphate dehydrogenase, an internal reference gene.

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
