# Peer review of "Induction of Chemerin on Autophagy and Apoptosis in Dairy Cow Mammary Epithelial Cells"

_animals, 2019, doi:10.3390/ani9100848_

Round 1

Reviewer 1 Report

The manuscript by Hu and collaborators tackles a very important question in the regulation of autophagy and apoptosis of bovine mammary epithelial cells. The paper presents several limitations:

There is a certain error in the expression of dairy cow mammary epithelial cells (DCMECs). Please modify it. The professional language is bovine mammary epithelial cells. References are too old. Did not find the literature of 2019 The background needs improvement in order to properly justify the work. The authors completely disregard the Chemerin-induced autophagy. It is indicated that the resuscitated mammary epithelial cells were used in vitro work, more details are needed to describe the DBMEC. Were the DBMEC characterized? It is unclear and poorly justified why the authors selected GAPDH as internal reference, authors should test at the least 4-5 reference genes and select the best one or two according to available algorythms, such as geNorm or the freely available NormFinder (see https://www.moma.dk/normfinder-software). Fluorescence quantitative PCR detection of expression: “The real-time PCR results were analyzed using a 2-△△ct model”:THIS POSITION SHOULD BE CITED. Figure 1 legends are simple graphic description instead of being information about how the experiments have been performed and what conclusions were got. This should be change as a figure legend should give a good understanding about how the experiment has been done. Please, provide the catalogue number for all the chemicals, reagents, antibodies, and assays used. The paper lacks a hypothesis. Please, state the hypothesis. Also, clearly indicated the objective(s) of the paper. Overall the discussion needs to improve and focus on the hypothesis (Fig. 6), and a description of the hypothesis (Fig. 6) should be written in the last paragraph, in Discussion

11.Language should be polished again.

Author Response

Dear Reviewer 1,

We feel great thanks for your professional review work on our manuscrip. We have studied your comments carefully and have made revision point by point. the revised place and changes is marked with red In this revised manuscript.Detailed revisions and explanations are listed and shown in author's notes files.

Sincerely,

Dr. Huixia Li

Reviewer 2 Report

The main question of this paper entitled  "Induction of Chemerin on autophagy and apoptosis in dairy cow mammary epithelial cells" submitted by  Bianhong Hu et al. study is to address whether chemerin induce autophagy and activate apoptosis in dairy cow mammary epithelial cells (DCMECs).

To answer this question the author measured the level of p62 and LC3  after treatment with Chemerin in DCMECs cells and they tested the apoptosis regulator Bax/Bcl2 and the downstream executioner of the cell death pathway,  cleaved caspase 3. 

The reasearch design is appropiate and to answer their questions they use different technical approach.

The authors should improve introduction that appear not well connected and they need to rephrase the suggested sentences.

The English language needs to be edited by a native English speaker.

Line 8: delete correspondence  (repeated two times)

Line 14:hormones and not hurmones

Line 15:  but it also involved

Line 24-26: rephrase

Line 36-37: rephrase

Line 49-50: apoptosis is a process of cell death

Methods

Line 106: add more information about the lysis buffer recipe used to collect the samples

Line 109: specify temperature

Line 119-120: rephrase and please specify how and whether you pre-treat the coverslip before to plate the cells

Line 132: add detail of binding buffer recipe

Section 2.6:  Describe how do you measure the average fluorescence intensity of the immunofluorescence staining

Figures

All the figures need to be consistent with the legends. Please make sure that you use the same character and same size

Figure 1 The size and the character of the pictures must to be the same. If you use capital letter in the figure be consistent in the legend. Increase the dimension of the number and characters used for the scale bars.

Figure 2.   Improve the quality of the pictures in panel A. In Chemerin treated cells, dapi signal looks over-saturated. Improve the quality of the LC3 staining in both control and Chemerin treatment in panel A. I also suggest to measure the % of cell positive for LC3 in addition to the average fluorescence intensity.  in Panel E add  the quantification of LC3 I.

Line 431 Specify the statistic test used in this analysis.

Figure 4. Panel AImprove the quality of the pictures, Dapi signal looks over-saturated and p62 signal looks not specific.

Author Response

Dear Reviewer 2,

We feel great thanks for your professional review work on our manuscrip. We have studied your comments carefully and have made revision point by point. the revised place and changes is marked with red In this revised manuscript.Detailed revisions and explanations are listed and shown in author's notes files.

Sincerely,

Dr. Huixia Li

Reviewer 3 Report

While this manuscript shows a small increase in autophagy componets with the treatment of mouse chemerin, the significance of this change is unknown.  The authors needed a positive control and the best would have been starvation of the cells for 24 h.  That way, a comparison of autophagy during starvation with and without chemerin would have been informative when compared to control and other various treatments.

Additionally, the authors used recombinant mouse chemerin and the active site on the mouse chemerin for the G-protein receptor (CMLKR1) is a 9-mer sequence.  It is interesting that mouse, human and bovine sequences in this region of CMLKR1 binding are significantly different.  What impact would the authors like to speculate for these differences?

Specifics:

Introduction - authors check your references dates - L 39 Li et al., 2018 is 2011 in References.  L 45 Mizushima et al., 2007 is 2011 in References.

Line 72 - add that this is mouse Chemerin.

L 119  Cells were inoculated onto a 24-w3ll sterile coverslip ?  Was this a coated coverslip as MECs will not adhere to glass.

L 132  What is "binding buffer"?

L 265-266  delete this sentence and associated figure 6.  

Author Response

Dear Reviewer 3,

We feel great thanks for your professional review work on our manuscrip. We have studied your comments carefully and have made revision point by point. the revised place and changes is marked with red In this revised manuscript.Detailed revisions and explanations are listed and shown in author's notes files.

Sincerely,

Dr. Huixia Li

Round 2

Reviewer 1 Report

Thanks the author for answering and solving all the questions. Experiments are meticulously planned and well executed. Based on the current situation of the article, I think this article can be published.

Reviewer 3 Report

We are still left with no real idea of what this small change means in the biology of chemerin and MEC.  The avoidance of any explanation of the use of mouse chemerin when bovine chemerin active receptor binding site is quite and significantly different.  The insistence upon the final figure only belongs in a review article.